# F4-ITS: Fine-grained Feature Fusion for Food Image-Text Search

## Abstract

The proliferation of digital food content has intensified the need for robust and accurate systems capable of fine-grained visual understanding and retrieval. In this work, we address the challenging task of food image-to-text matching, a critical component in applications such as dietary monitoring, smart kitchens, and restaurant automation. We propose F4-ITS: Fine-grained Feature Fusion for Food Image-Text Search, a training-free, vision-language model (VLM)-guided framework that significantly improves retrieval performance through enhanced multimodal feature representations. Our approach introduces two key contributions: (1) a uni-directional(and bi-directional) multi-modal fusion strategy that combines image embeddings with VLM-generated textual descriptions to improve query expressiveness, and (2) a novel feature-based re-ranking mechanism for top-k retrieval, leveraging predicted food ingredients to refine results and boost precision. Leveraging open-source image-text encoders, we demonstrate substantial gains over standard baselines - achieving $\sim$10% and $\sim$7.7% improvements in top-1 retrieval under dense and sparse caption scenarios, and a $\sim$28.6% gain in top-k ingredient-level retrieval. Additionally, we show that smaller models (e.g., ViT-B/32) can match or outperform larger counterparts (e.g., ViT-H, ViT-G, ViT-bigG) when augmented with textual fusion, highlighting the effectiveness of our method in resource-constrained settings. Code and test datasets will be made publicly available.

## 1 Introduction

Existing image-text models like CLIP Radford et al. (2021) have demonstrated remarkable capabilities in aligning image and text embeddings across a vast range of general domains. However, their performance often diminishes when confronted with the extreme fine-grained distinctions required in specialized domains like food. The core challenge lies in bridging the semantic gap between visual appearance and the precise textual nuances that define a specific dish or its components. For instance, distinguishing between "Chicken Curry with Basmati Rice" and "Lamb Curry with Jeera Rice" demands recognition of subtle meat and rice grain differences, alongside an understanding of specific ingredient names.

Current methods often provide a fine-tuning strategy where image-text models(such as CLIP (Radford et al., 2021), SigLIP (Zhai et al., 2023)) are trained on in-domain semantically rich text descriptions or combine vision-language models in a zero-shot manner to bridge this semantic gap. However, there are two problems with these approaches: 1. The manual cost of creating such rich food descriptions is quite high. 2. The accuracy of food description and ingredient caption retrieval still requires a good amount of improvement as fusion of image and text features along with precise ranking of food ingredients remain a challenging task.

This paper proposes a novel training-free system designed to overcome these challenges, focusing on two key sub-problems: retrieving the single best holistic food description for a given food image, and retrieving the top-k most relevant food ingredients. Our solution integrates image-text models and the powerful VLMs and introduces two primary novelties to enhance the retrieval performance:

1. Uni and bi-directional multi-modal embedding fusion: We introduce a bi-directional fusion(in addition to uni-directional) between the raw image embedding and a VLM-generated "dense/rich

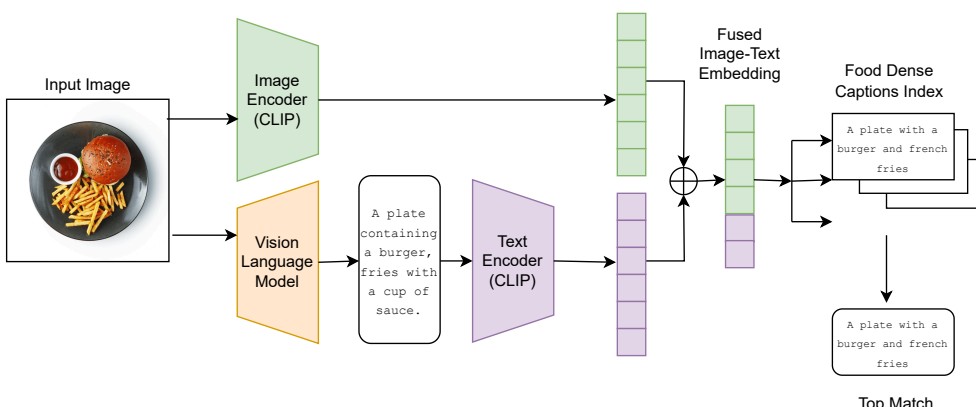

Figure 1: Overview of the proposed F4-ITS: image-text feature fusion architecture for fine-grained food image-text search For dish understanding. Given a set of food captions index and a query image to match against, our system makes use of image-text models(CLIP) and VLMs together to generate enhanced image and text representations which improves the overall retrieval.

food description" embedding. The fusion with the food descriptions bridges this semantic gap and allows the system to capture the fine-grained, subtle food details. We show how this fusion strategy helps smaller models(like ViT-B) perform on par or even better than the larger models(ViT-H, ViT-G, ViT-bigG) enabling accurate real-time search systems. Our paper also presents thorough comparison of using dense and sparse food captions and how having high signal captions improve the overall search accuracy.

2. Feature re-ranking with ingredient-level embeddings: For top-k retrieval of food ingredients, we propose a novel feature based re-ranking strategy. After an initial retrieval using a fused embedding, we break down the VLM-generated "sparse food ingredient description" into individual food item embeddings. These fine-grained item embeddings are then used to re-score the top-k candidates, identifying the most relevant captions by focusing on specific ingredient or dish component matches.

## 2 RELATED WORK

CLIP Radford et al. (2021) and SigLIP Zhai et al. (2023) models are generally trained to align image and text features, thereby enabling cross-modal retrieval(image-to-text as well as text-to-image). However, the performance of these models are highly dependent on the training data distribution - both the image and the captions. Most of these pre-trained models are trained on text descriptions with high level concepts or coarse-grained(eg: rice with meat on a plate) and not at a fine-grained level(eg: a plate with fried rice and chicken). The need for fine grained categorization has led to the emergence of two major research directions: 1. Training based and 2. Zero-shot/training-free methods that, on a high level, aim to better align image features with highly informative and semantically rich textual descriptions.

Training based approaches such as ZS-CTIR Liu et al. (2024), DistillCLIP Csizmadia et al. (2025), ADEM-VL Hao et al. (2024) and Everything can be described in words Bi & Xu (2025) explore the image-text alignment problem using various supervised or distillation-based techniques. While training-free methods such as PDV Tursun et al. (2025), TF-ZS-CIR Wu et al. (2025) has shown that image-to-image retrieval can be enhanced by augmenting CLIP features with VLMs(through averaging or weighted sum or concatenation), demonstrating how merging of image features with text features can direct embedding towards the target features.

Vision-Language Models (VLMs) (Alayrac et al., 2022; Dai et al., 2023) have demonstrated strong performance on tasks such as visual question answering and object recognition. However, while they may not be inherently optimized for image-text retrieval, their strong generalization ability in recognizing natural images, particularly food, makes them a valuable component in search systems.

Our paper proposes a training-free zero-shot framework to not just retrieve(through uni and bi-directional fusion) but also re-rank fine-grained textual descriptions(individual ingredients) from images through VLM guidance. Although our method makes use of CLIP and VLMs features together as in the past methods, we differ from the problem domain of fine-grained food categorization where highly informative captions matter more(than sparse captions which is experimentally proven) and devise a bi-directional fusion and re-ranking algorithm for more precise image to text search.

## 3 PROBLEM FORMULATION

Given a query food image $I$ and an index or retrieval corpus $C$ consisting of food-related textual descriptions, the objective is to retrieve a ranked list of captions from $C$ that are most semantically and visually relevant to the contents of $Q$. This task is particularly challenging in the food domain due to the presence of subtle visual distinctions across dishes, variations in ingredients, and differing levels of textual description granularity.

To address the need for both holistic dish understanding and fine-grained ingredient recognition, we decompose the overall problem of food image-to-text retrieval into two complementary sub-tasks, each tailored for a specific downstream purpose:

1. **Single Image-Text Retrieval (Dense Caption Retrieval):** This task involves retrieving the single most descriptive and semantically aligned caption for a given food image. The caption set in this case consists of dense, highly informative textual descriptions that capture not just the type of dish, but also detailed attributes such as preparation style, presentation, and accompanying ingredients. *Example:* `"Hearty bowl of savory braised pork belly, tender glass noodles, and green vegetables."` The goal is to retrieve the top-1 caption that best captures the full semantics of the visual input.

2. **Top-$k$ Image-Text Retrieval (Sparse Ingredient Retrieval):** In this task, the captions are sparse and correspond to individual food ingredients or components that are visually present in the image. The aim is to retrieve the top-$k$ most relevant ingredient-level captions from the index, capturing all (or most) of the distinct food items seen in the image. *Example:* `"black beans, corn, bell pepper, tomato."` This task allows evaluation of ingredient-level precision and is particularly important for applications such as dietary tracking or nutritional analysis.

## 4 APPROACH

Our proposed multi-modal search system consists of three core components:

1. Fine-grained embedding extraction using image-text models.

2. Uni and bi-directional image-text feature fusion.

3. Feature re-ranking based on fine-grained food ingredients.

The process of extracting image representations + feature fusion with VLM textual description followed by retrieval on captions remains common for both the sub-tasks(Figure 1 and 3). Only difference is in the type of captions(dense/sparse) used for these tasks. Additionally, for the top-k retrieval task(Figure 3), we perform sparse text feature based re-ranking on the originally retrieved results. Following sub-sections talks in detail about the approaches proposed for both the sub-tasks.

**Definitions:**
*Encoders: Image Encoder(IE) and Text Encoder(TE).*
*Index: Dense Caption Index(DCI) and Sparse Caption Index(SCI)*, used for single and top-k retrieval tasks respectively.

### 4.1 EMBEDDING EXTRACTION

The system makes use of the pre-trained large-scale image-text models as the base image and text embedding extractors. Given an index with a list of captions C, extract the text embeddings using

the Text Encoder TE. These embeddings form our searchable database. In case of bi-directional fusion, these embeddings are modified by fusing with the query image embedding at runtime.

## 4.2 Single Image-Text Retrieval with Multi-modal Fusion

In a single image-text retrieval setting, the objective is to retrieve a dense, rich food description/caption(from the dense caption index DCI) for a given query image. An image is of a dish with multiple food items and a caption can be "The plate contains grilled chicken with potatoes and garlic sauce."

- **Image Embedding Extraction**: For the given query image $IQ$, we extract its image embedding using the Image Encoder IE.

$$E_{img} = ImageEncoder(IQ) \tag{1}$$

- **VLM generated rich item description**: Utilizing a VLM, we generate a rich food dish description *DenseCaptionText*, that is dense enough to capture various attributes of the food dish - including the ingredients and arrangement of these ingredients in the image.

$$E_{densecaption} = TextEncoder(DenseCaptionText) \tag{2}$$

- **Image-Text Feature Fusion**: We use a simple weighted sum fusion of image and text embeddings. We set the weights w_img=0.7 and w_text=0.3, obtained through extensive experiments.

$$E_{fused} = w_{img} * E_{img} + w_{text} * E_{densecaption} \tag{3}$$

The above represents a uni-directional fusion where only the query representation is improved. Another enhancement is to fuse the index embeddings(DCI and SCI) with the query image embedding(bi-directional).

$$E_{Ci fused} = w_{img} * E_{img} + w_{text} * E_{C_i} \tag{4}$$

Here, the weights used are w_img=0.3 and w_text=0.7.

- **Retrieval**: The system retrieves the best matching food description from the dense caption index, DCI by computing cosine similarity for the fused embedding $E_{fused}$ against all the text embeddings in the index.

$$C_{best} = argmax(cosine\_similarity(E_{fused}, E_{Ci})) \tag{5}$$

* We can use either uni-directional fusion or bi-directional fusion during the retrieval. For simplicity, we show only uni-directional fusion based retrieval in the remainder of the paper.

## 4.3 Top-k Retrieval with Feature Re-ranking

In a top-k image-text retrieval setting, the objective is to retrieve the most relevant yet diverse individual food ingredient captions(from the sparse caption index SCI) for a given query image. An image usually is of a dish with several food items and a caption can be "french toast", "coffee".

- **Image Embedding Extraction**: For the given query image IQ, we extract its image embedding using the Image Encoder IE.

$$E_{img} = ImageEncoder(IQ) \tag{6}$$

- **Initial top-k retrieval**: First step in the top-k retrieval involves fusing the image embedding $E_{img}$ with the sparse caption embedding $E_{sparsecaptionVLM}$, generated by querying a VLM asking for a "sparse item description"(only the individual food ingredients present in the image). This description is concise, focusing on key entities or prominent components (e.g., "chicken, rice, curry leaves") rather than a verbose sentence.

$$E_{sparsecaption} = TextEncoder(SparseCaptionText) \tag{7}$$

We stick to the weights(w_img=0.7, w_text=0.3) for fusion as in top-1 retrieval.

$$E_{fused} = w_{img} * E_{img} + w_{text} * E_{sparsecaption} \tag{8}$$

Finally, we retrieve the top k matches using cosine similarity to form the candidate set $C_{TopN} = C_1, C_2, ..., C_N$.

$$C_{TopN} = topk(cosine\_similarity(E_{fused}, E_{caption_i})) \tag{9}$$

- **Reranking through VLM generated item level descriptions**: This represents our second primary contribution, designed to precisely differentiate highly similar dishes within the candidate set $C_{TopN}$.

  - **Fine-Grained Item Extraction**: We take the VLM-generated "sparse item description" (e.g., "chicken, rice, curry leaves") and parse it into individual food item phrases, *ParsedItems*. For each extracted phrase Pj (e.g., "chicken," "rice," "curry leaves"), we generate a separate item-specific embedding.

$$E_{P_j} = TextEncoder(P_j) \tag{10}$$

  - **Max-Similarity Reranking**: For each candidate caption $Ck \in CTopN$, we re-evaluate its relevance by computing its maximum cosine similarity with all of the individual food item embeddings $E_{P_j}$. This strategy ensures that captions containing any specific, strong matches to the identified food items in the image are highly scored, even if other parts of the caption are less aligned.

$$ScoreRerank(C_k) = \max_{P_j \in ParsedItems} cosine\_similarity(E_{C_k}, E_{P_j}) \tag{11}$$

  This approach allows for highly discriminative semantic alignment at a granular level. For example, if the image clearly depicts "shrimp" and a candidate caption mentions "shrimp," it will receive a high reranking score, effectively distinguishing it from a visually similar dish mentioning "chicken."

  - The captions in $C_{TopN}$ are then re-ordered based on their $ScoreRerank(C_k)$ values, and the top-k (e.g., k=5 or k=10) captions from this refined list are presented as the final search result. This hierarchical reranking significantly boosts precision by leveraging the fine-grained information extracted from the VLM outputs, which is crucial for distinguishing between subtle food variations.

## 5 DATASET

VLM Metafood Challenge MetaFood25 (2025) focuses on the problem of cross modal retrieval(image to text captions) and provides two datasets - MTF25-VLM-Challenge-Web Rodríguez-de-Vera (2025b) and MTF-25-VLM-Challenge-Synth Rodríguez-de-Vera (2025a), with 139K and 258K image and rich food caption pairs respectively. For the purpose of benchmarking, we take a small subset of this dataset(13K and 15K image-caption pairs from Synth and Web splits) and perform evaluation under both single image-text and top-k retrieval settings. We present two evaluation datasets - MTF25-VLM-Challenge-Dataset-Web-13K dataset consists of 12,680 images and caption pairs and MTF25-VLM-Challenge-Dataset-Web-15K dataset consists of 15127 images and caption pairs. Every image is associated with a densely rich and a sparse caption. We use synthetically generated captions as ground truth to circumvent the problem of noisy captions and for source-target caption alignment.

## 6 EXPERIMENTS

We perform thorough experiments across different datasets, across various caption types(dense and sparse) and fusion weights. We use OpenCLIP MLFoundations (2023) pre-trained models as our base image-text encoders.

## 6.1 SINGLE IMAGE-TEXT RETRIEVAL

The objective of this experiment(Table 1 and 4) is to evaluate how well our system is able to retrieve top-1 and top-5 food captions that best describes the image. Under single image-text retrieval, we evaluate several pretrained open-source image-text models: CLIP, SigLIP and their fused counterparts. We maintain two types of ground truth captions - a. Dense index captions(DCI) and b. Sparse index captions(SCI). For each GT caption type, we perform fusion for dense as well sparse text features(prediction) generated with the help of Gemini-2.5-FlashTeam (2025a)/Gemma-3nTeam (2025b) and CLIP/SigLIP text encoders.

Under this experimental setting, we use $w\_img = 0.7$ and $w\_text = 0.3$ as fusion parameters(after extensive experiments). Evaluation metric used are Recall@1(top-1, exact match) and Recall@5.

Note: Although the original problem setup involves matching with only dense captions(eg: *"Vibrant chicken salad with crisp greens, fresh tomatoes, and sweet bell peppers, garnished with olives on a rustic table"*), we also benchmark for matching with sparse captions(eg: *"chicken, salad, lettuce, tomato, bell pepper, olives"*) to understand retrieval performance under different information density.

Table 1: Dense index: Recall@1 and Recall@5 for single image-text retrieval on MTF25-VLM-Challenge-Dataset-Web-13K and Synth-15K Datasets. Pretrained OpenCLIP variants and SigLIP are evaluated using both dense and sparse captions(prediction). Bold indicates that feature fusion significantly improves over the baseline, and underline highlights the overall best R@1 and R@5 across all evaluated models.

| Model | MTF25-VLM-Web-13K | | | | MTF25-VLM-Synth-15K | | | |
| --- | --- | --- | --- | --- | --- | --- | --- | --- |
| | Dense | | Sparse | | Dense | | Sparse | |
| | R@1 | R@5 | R@1 | R@5 | R@1 | R@5 | R@1 | R@5 |
| ViT-B-32 (baseline, w/o fusion) | 0.335 | 0.581 | - | - | 0.339 | 0.630 | - | - |
| F4-ITS(+ Gemma-3n) | 0.444 | 0.689 | 0.385 | 0.636 | 0.438 | 0.707 | **0.399** | **0.673** |
| F4-ITS(+ Gemini) | **0.512** | **0.752** | **0.420** | **0.673** | **0.449** | **0.724** | 0.384 | 0.664 |
| ViT-g-14 | 0.451 | 0.702 | - | - | 0.442 | 0.724 | - | - |
| F4-ITS(+ Gemma-3n) | 0.538 | 0.776 | 0.488 | 0.736 | 0.513 | 0.771 | **0.476** | **0.742** |
| F4-ITS(+ Gemini) | **0.587** | **0.813** | **0.527** | **0.772** | **0.525** | **0.783** | 0.472 | 0.743 |
| ViT-bigG-14 | 0.442 | 0.688 | - | - | 0.415 | 0.698 | - | - |
| F4-ITS(+ Gemma-3n) | 0.468 | 0.711 | 0.453 | 0.696 | **0.429** | 0.698 | **0.430** | **0.701** |
| F4-ITS(+ Gemini) | **0.499** | **0.730** | **0.468** | **0.713** | 0.428 | **0.705** | 0.414 | 0.691 |
| ViT-H-14-378-quickgelu | 0.519 | 0.761 | - | - | 0.498 | 0.770 | - | - |
| F4-ITS(+ Gemma-3n) | 0.568 | 0.790 | 0.512 | 0.751 | 0.537 | 0.794 | **0.496** | **0.761** |
| F4-ITS(+ Gemini) | **0.609** | **0.820** | **0.534** | **0.771** | 0.547 | **0.801** | 0.481 | 0.747 |
| ViT-L-16-SigLIP2 | 0.525 | 0.759 | - | - | 0.488 | 0.758 | - | - |
| F4-ITS(+ Gemma-3n) | 0.560 | 0.783 | 0.455 | 0.692 | 0.507 | 0.765 | **0.423** | **0.689** |
| F4-ITS(+ Gemini) | **0.586** | **0.80** | **0.466** | **0.702** | **0.507** | 0.763 | 0.395 | 0.658 |

## 6.2 TOP-K RETRIEVAL

The objective of this experiment(Table 2) is to evaluate how well our system can retrieve and rank individual ingredients from food images. Similar to single image-text retrieval setting, we evaluate pre-trained image-text models along with the fused variants, but only with an index of sparse captions(SCI) containing individual ingredients. We maintain one GT sparse caption index and perform fusion using sparse text features(prediction) generated with the help of Gemini-2.5-Flash/Gemma-3n and CLIP/SigLIP text encoders.

Under this experimental setting, we use *w_img = 0.7* and *w_text = 0.3* as fusion parameters(after extensive experiments). Evaluation metric used is mAP. The value of k varies for each image and is decided based on the number of items returned in the GT sparse caption. For eg: "scallop, cauliflower, greens, herb oil" will have k=4.

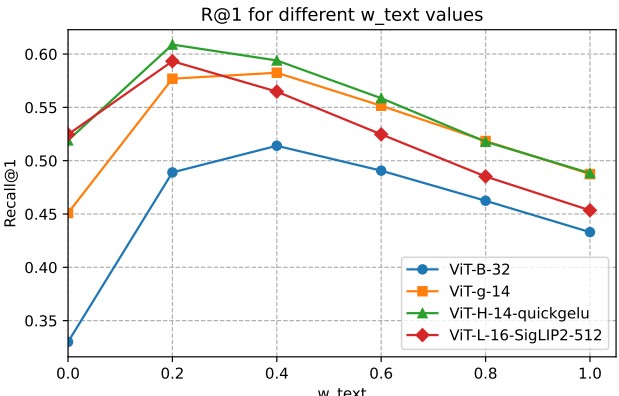

Figure 2: Feature fusion accuracy over different *w_text* values. Over fusion(and no fusion) of text features leads to accuracy drop, while 20-30% text fusion guides the model towards better retrieval.

# 7 RESULTS

## 7.1 SINGLE IMAGE-TEXT RETRIEVAL

Weighted image-text fusion shows significant increase in accuracy when fused with VLM textual descriptions - for both Gemini-2.5-Flash as well as Gemma-3n. We observe a minimum increase(in Recall@1) of 5.7%(ViT-bigG-14) and maximum increase of 17.7%(ViT-B-32) for DCI(Table 1) and a min and max increase of 3.4%(ViT-big-14) and 14.1%(ViT-B-32) respectively, for SCI(Table 4). Heterogeneous matching of DCI with sparse caption predictions and SCI with dense caption predictions does not provide a major boost(as compared to the homogeneous matching) - 1.1% to 3.5%.

One sharp observation is that, smaller models perform on par or sometimes even better than the larger ones in case of fusion. For eg: In Table 1, ViT-B-32 shows a top-1 accuracy of 51.2% while ViT-bigG-14 stands at 49.9%. Our framework shows that the performance of these models can improve without any finetuning by simply adding more information produced by another powerful model. This also helps in deploying smaller models that are as accurate as the larger models but can run at a much lower compute.

Another key observation is that, dense captions are much more rich in signal(which is straightforward) and works better than using sparse captions. Figure 4. shows that feature fusion with dense captions improves over sparse captions by an average of 8%, ranging from 3.7%(min) to 14.3%(max) difference for Recall@1.

We see that more weighting for text for smaller models and less weighting for larger models is optimal(Table 1 and 4) as the observation is that these larger models(ViT-L, ViT-H and ViT-bigG) are already well aligned and able to capture the fine-grained pattern differences to a reasonable extent. Figure 2 shows how over fusion(or no fusion of text features) causes accuracy drop while moderate fusion(at 0.2-0.3) leads to optimal fusion.

For synthetic datasets, we see that the gains due to fusion are not as high as real images. This can be attributed to the shift in training distribution as these open-source image-text models are in general trained on large-scale real-world web image datasets.

We also benchmark F4-ITS on the original noisy captions provided and notice that the fusion works(not as effective as for clean captions) reasonably okay in case of noisy captions(Table 3). Although the gains are 1-2%, our hypothesis of using VLM guidance for enhanced image representation stay true.

Table 2: Sparse index: mAP for top-k image-text retrieval on MTF25-VLM-Challenge-Dataset-Web-13K and Synth-15K Datasets. Bold indicates that feature re-ranking significantly improves over the baseline, and underline highlights the overall best mAP across all evaluated models.

| Model | MTF25-VLM-Web-13K | MTF25-VLM-Synth-15K |
|---|---|---|
| | mAP | mAP |
| ViT-B-32 (baseline) | 0.084 | 0.091 |
| F4-ITS(+ Gemma-3n) | 0.277 | 0.334 |
| F4-ITS(+ Gemini) | **0.379** | **0.378** |
| ViT-g-14 | 0.064 | 0.074 |
| F4-ITS(+ Gemma-3n) | 0.264 | 0.340 |
| F4-ITS(+ Gemini) | **0.366** | **0.393** |
| ViT-bigG-14 | 0.104 | 0.102 |
| F4-ITS(+ Gemma-3n) | 0.284 | 0.330 |
| F4-ITS(+ Gemini) | **0.380** | **0.375** |
| ViT-H-14-378-quickgelu | 0.106 | 0.113 |
| F4-ITS(+ Gemma-3n) | 0.269 | 0.293 |
| F4-ITS(+ Gemini) | **0.362** | **0.335** |
| ViT-L-16-SigLIP2-512 | 0.136 | 0.170 |
| F4-ITS(+ Gemma-3n) | 0.286 | 0.364 |
| F4-ITS(+ Gemini) | **0.374** | **0.407** |

Finally, our bi-directional fusion strategy doesn't really improve or decrease the performance over the uni-directional fusion(where only the query is modified). It is also optimal to use uni-directional for performance reasons.

### 7.2 TOP-K IMAGE-TEXT RETRIEVAL

Results for top-k retrieval shows significant improvements(especially in search precision) of upto 28.6%(mAP) over the traditional system without re-ranking(Table 2). Using feature re-ranking ensures that the top search results include the most relevant ones. Our feature re-ranking system, in a way, acts as a high-recall -> high precision search system, leading to more user relevant search results.

## 8 INTUITION

### 8.1 WHY DOES THE PROPOSED IMAGE-TEXT FEATURE FUSION WORK?

Image-text feature fusion has been well discussed in the literature where weighted sum, average and concatenation are widely used strategies(training-free). There are also several training-based methods innovating in cross attention across image and text modalities. The proposed feature fusion strategy works because of target aligned text descriptions and usage of highly accurate VLMs for description generation. The more the descriptions are aligned with the target index captions and the more accurate they are, the better the fusion is(zero-shot).

Table 3: Noisy (dense) index: Recall@1 and Recall@5 for single image-text retrieval on MTF25-VLM-Challenge-Dataset-Web-13K. Numbers indicate that the noisy captions lead to little to no improvement due to feature fusion(sometimes even lower than the baseline). Fusion parameters - w_img: 0.95 and w_text: 0.05

| Model | MTF25-VLM Web 13K | |
| | R@1 | R@5 |
| --- | --- | --- |
| ViT-B-32 (baseline) | 0.341 | 0.582 |
| F4-ITS(+ Gemini) | **0.359** | **0.594** |
| ViT-L-14 | 0.427 | **0.672** |
| F4-ITS(+ Gemini) | **0.431** | 0.668 |
| ViT-H-14 | **0.466** | **0.715** |
| F4-ITS(+ Gemini) | 0.466 | 0.714 |
| ViT-g-14 | 0.438 | **0.685** |
| F4-ITS(+ Gemini) | **0.445** | 0.685 |
| ViT-bigG-14 | **0.497** | **0.744** |
| F4-ITS(+ Gemini) | 0.495 | 0.741 |
| ViT-H-14-378-quickgelu | 0.523 | **0.763** |
| F4-ITS(+ Gemini) | **0.527** | 0.759 |
| ViT-L-16-SigLIP2-512 | **0.492** | **0.732** |
| F4-ITS(+ Gemini) | 0.491 | 0.732 |

## 8.2 WHY DENSE CAPTIONS ARE BETTER THAN SPARSE CAPTIONS?

Dense captions usually carry high signal(adding to the already present food items) compared to their sparse counterpart. In addition to this, it also aligns well with how these foundation models like OpenCLIP, SigLIP are trained - noisy web captions which are high in information. High signal and alignment with training distribution helps in more accurate image-text search.

## 9 F4-ITS AS A GENERAL PURPOSE IMAGE SEARCH SYSTEM

The paper proposes a framework for a training-free high accuracy image search - not just applicable to image-to-text, but also to text-to-image and image-to-image search applications. The fusion of image and text modalities guided by VLMs offers an effective way for fine-grained instance recognition, while the reranking block(in text modality) helps refine the search results better, thereby improving the overall precision. Although this paper focuses on a narrow domain of food image search, the proposed system is much more general and can be applied to various domains including retail etc.

## 10 CONCLUSION

We presented a novel training-free methodology for accurate food image-text search at a fine-grained level. We observe that by efficiently fusing the image features from image-text models and text features from VLM, we can extract significant gains in retrieval accuracy. A future scope of work is adaptive fusion of image and text features, where the weights can be determined dynamically. We believe this work can be applied in real world multi-modal search systems and improve the quality of search results and thereby user experience.

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

# A APPENDIX

## A.1 F4-ITS RE-RANKING ARCHITECTURE

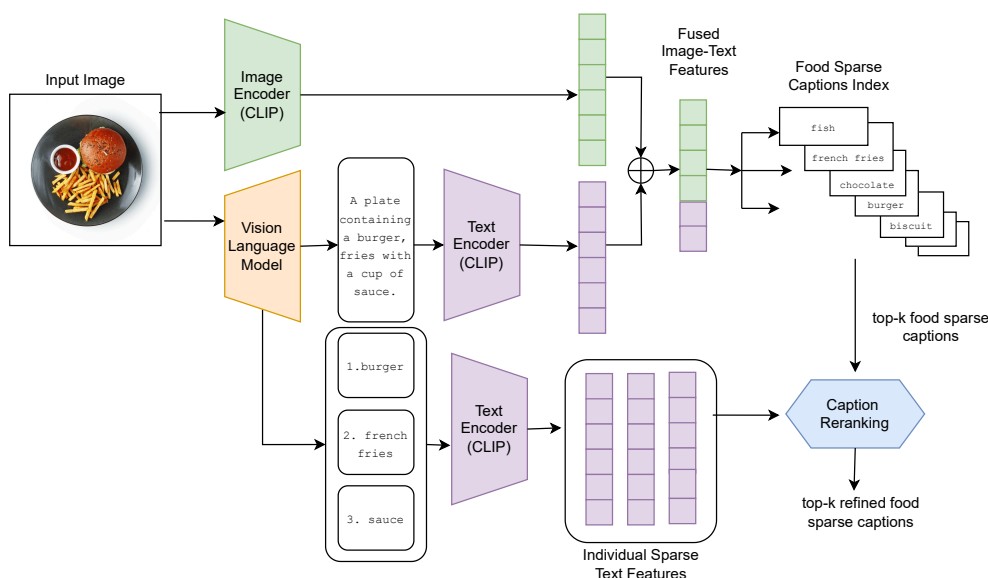

Figure 3: Overview of feature fusion + re-ranking architecture for fine-grained image-food ingredient Search. Given a set of sparse caption index(food ingredients), our system uses the fusion architecture followed by a feature re-ranker that improves relevance of search results.

## A.2 FEATURE FUSION - DENSE VS SPARSE CAPTIONS

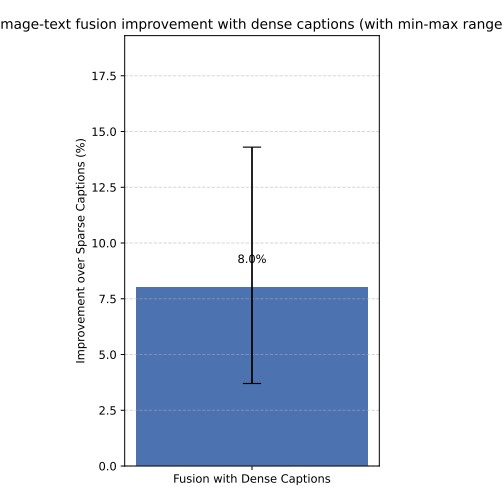

Figure 4: Feature Fusion with dense captions improves over sparse captions by an average of 8%, ranging from 3.7% to 14.3%.

## A.3 EXPERIMENT RESULT: SINGLE IMAGE-TEXT RETRIEVAL WITH SPARSE INDEX

Table 4: Sparse index: Recall@1 and Recall@5 for single image-text retrieval on MTF25-VLM-Challenge-Dataset-Web-13K and Synth-15K Datasets. Pretrained OpenCLIP variants and SigLIP are evaluated using both dense and sparse captions(prediction). Bold indicates that feature fusion significantly improves over the baseline. Underline highlights the overall best R@1 and R@5 across all evaluated models.

| Model | MTF25-VLM-Web-13K | | | | MTF25-VLM-Synth-15K | | | |
| | Dense | | Sparse | | Dense | | Sparse | |
| | R@1 | R@5 | R@1 | R@5 | R@1 | R@5 | R@1 | R@5 |
|---|---|---|---|---|---|---|---|---|
| ViT-B-32 | | | | | | | | |
| (baseline, w/o fusion) | 0.205 | 0.416 | - | - | 0.232 | 0.468 | - | - |
| F4-ITS(+ Gemma-3n) | 0.294 | 0.525 | 0.332 | 0.554 | 0.306 | 0.556 | 0.341 | 0.590 |
| F4-ITS(+ Gemini) | **0.346** | **0.581** | **0.406** | **0.632** | **0.314** | **0.576** | **0.354** | **0.598** |
| ViT-g-14 | 0.294 | 0.530 | - | - | 0.306 | 0.568 | - | - |
| F4-ITS(+ Gemma-3n) | 0.364 | 0.605 | 0.380 | 0.615 | 0.363 | 0.626 | 0.386 | 0.635 |
| F4-ITS(+ Gemini) | **0.410** | **0.651** | **0.448** | **0.677** | **0.375** | **0.638** | **0.401** | **0.651** |
| ViT-bigG-14 | 0.305 | 0.541 | - | - | 0.306 | **0.563** | - | - |
| F4-ITS(+ Gemma-3n) | 0.324 | 0.556 | 0.387 | 0.623 | 0.301 | 0.560 | 0.377 | 0.634 |
| F4-ITS(+ Gemini) | **0.339** | **0.576** | **0.441** | **0.668** | **0.305** | 0.560 | **0.389** | **0.636** |
| ViT-H-14-378-quickgelu | 0.350 | 0.589 | - | - | 0.365 | 0.638 | - | - |
| F4-ITS(+ Gemma-3n) | 0.396 | 0.637 | 0.410 | 0.642 | 0.392 | 0.657 | 0.408 | 0.670 |
| F4-ITS(+ Gemini) | **0.428** | **0.670** | 0.466 | 0.688 | **0.402** | **0.665** | 0.413 | 0.665 |
| ViT-L-16-SigLIP2 | 0.337 | 0.562 | - | - | 0.331 | 0.596 | - | - |
| F4-ITS(+ Gemma-3n) | 0.377 | 0.601 | 0.395 | 0.621 | 0.345 | 0.598 | 0.378 | **0.630** |
| F4-ITS(+ Gemini) | **0.390** | **0.620** | **0.433** | **0.658** | **0.352** | **0.604** | **0.385** | **0.630** |

