# OpenReview forum: "F4-ITS: Fine-grained Feature Fusion for Food Image-Text Search"
_ICLR.cc/2026/Conference — Submitted to ICLR 2026_

### Official Review · Reviewer_2FP2 · 2025-10-28

**Soundness:** 1
**Presentation:** 1
**Contribution:** 1
**Rating:** 2
**Confidence:** 5

**Summary:**

The paper proposes F4-ITS (Fine-grained Feature Fusion for Food Image-Text Search), a training-free framework designed to improve fine-grained cross-modal retrieval in the food domain. The method fuses image embeddings from pretrained CLIP-like models with dense or sparse textual embeddings generated by large vision-language models (VLMs) such as Gemini or Gemma.
Evaluations on the MetaFood25 dataset show moderate gains across multiple ViT backbones.

The technical novelty is minimal—the method mainly combines weighted feature fusion and cosine-based re-ranking, both well-explored in prior zero-shot retrieval works. Experiments are limited to a single dataset without generalization or robustness analysis, and key design choices (e.g., fusion weights) lack justification. Overall, the contribution is incremental and insufficient for a top-tier venue.

**Strengths:**

- Applicable across ViT-B to ViT-bigG models; findings (e.g., smaller models benefit more) are interesting and practically valuable.

- The experiments are clear and the results consistently show measurable improvements (e.g., +10% Recall@1, +28.6% mAP), demonstrating the empirical effectiveness of simple feature fusion under the food domain.

**Weaknesses:**

- Technical novelty is very limited. The approach primarily relies on a weighted sum fusion of image and VLM-generated text embeddings, followed by cosine-based re-ranking — both are straightforward extensions of existing zero-shot fusion and retrieval strategies.

- The paper repeatedly fixes fusion weights, there is no justification or ablation for the setting of $w_{img}$ or $w_{text}$. Similarly, other design choices (e.g., selection of Gemini/Gemma captions, reranking threshold) are not thoroughly justified.

- Insufficient experiments. Experiments are restricted to MetaFood25; there is no evaluation on broader or unseen datasets such as Food-101 or Recipe1M. This limits the claimed generalization. No direct comparison with prior zero-shot or fine-grained food retrieval methods (e.g., fine-tuned CLIP variants) is presented.

- The method heavily relies on VLM-generated dense/sparse captions, which might not generalize across domains. There’s no discussion of robustness under noisy or biased caption generation.

- While the authors claim the system can generalize to other image-text retrieval domains, all evidence is food-specific, and the method lacks evaluation or discussion supporting such transferability. The generality of the proposed method is over-claimed.

**Questions:**

Please compare your results with the existing state-of-the-art works in the food retrieval task.

---

### Official Review · Reviewer_8vMY · 2025-10-30

**Soundness:** 2
**Presentation:** 1
**Contribution:** 1
**Rating:** 2
**Confidence:** 4

**Summary:**

This paper proposes F4-ITS, a training-free framework designed to improve fine-grained food image-to-text retrieval. The authors identify that general-purpose models like CLIP struggle with subtle distinctions in specialized domains like food. The proposed solution has two main components:
- A common weighted sum multi-modal fusion strategy that combines a standard image embedding (from CLIP, etc.) with a text embedding of a "dense" description generated by a VLM, like Gemini 2.5.
- A common feature-based re-ranking mechanism for top-k ingredient retrieval, where VLM-generated "sparse" ingredient lists are used to re-score an initial set of candidates based on maximum similarity.

**Strengths:**

-  The results are promising. The authors show that with the weighted sum fusion methods, it improve ~28.6% in top-k ingredient retrieval and ~10% in desnse caption retrieval.

**Weaknesses:**

- Lack of novelty: The paper's main weakness is its limited novelty. The first key contribution is a weighted sum fusion method. This is a widely-known, basic ensemble technique [1]. The paper attempts to differentiate itself by focusing on the food domain, but this does not constitute a novel algorithmic contribution.
- Unclear results: The table 1, 2, and subsequent tables do not specify the types of fusion methods (uni/bi direction fusion) employed to obtain the results.
- Missing details: What's the prompt do you use for the Gemini and Gemma?
- The comparisons are weak. In Table 1 and Table 2, the F4-ITS method is only compared against the baseline, w/o fusion. This is a weak argument. The authors should have compared their weighted-sum approach against other simple fusion methods mentioned in the related work, such as simple averaging or concatenation. More importantly, they didn't compare against any other training-free retrieval methods from related work, like PDV.

[1] Liang, Paul Pu, Amir Zadeh, and Louis-Philippe Morency. "Foundations & trends in multimodal machine learning: Principles, challenges, and open questions." ACM Computing Surveys 56.10 (2024): 1-42.

**Questions:**

See weaknesses.

---

### Official Review · Reviewer_s3F6 · 2025-10-31

**Soundness:** 2
**Presentation:** 2
**Contribution:** 2
**Rating:** 2
**Confidence:** 4

**Summary:**

This paper proposes the F4-ITS, a framework for Food Image-Text Search. This approach tackles two problems: 1) Single Image-Text Retrieval (Dense Caption Retrieval) and 2) Top-k Image-Text Retrieval (Sparse Ingredient Retrieval). The main idea is to fuse image and image caption features, and then retrieve the food text. The authors performed experiments on Food datasets and evaluated the framework's performance using different ViT architectures.

**Strengths:**

This paper tackles an important task: Food Image-Text retrieval. This is important for downstream applications such as dietary monitoring, nutritional analysis, and so on.

The framework is easy to understand.

**Weaknesses:**

There is no technical innovation in the framework. Using CLIP image and text encoders to extract image and text features is widely used, and fusing them using weights is well-known.

The CLIP model is relatively small. Larger models, such as BLIPv3, Qwen2.5-VL, should be used to test the performance of the proposed framework.

**Questions:**

It's better to explain what the "Dense Caption Index" is. I guess it's the input text information.

---

### Official Review · Reviewer_QYUf · 2025-11-01

**Soundness:** 2
**Presentation:** 2
**Contribution:** 2
**Rating:** 2
**Confidence:** 4

**Summary:**

This paper presents F4-ITS, a training-free framework for food image-text retrieval that  mixes CLIP/SigLIP with vision-language models to link food pictures and words. It fuses features both one-way and two-way, then re-ranks answers for a better match. Experiments are conducted on MetaFood data to show its effectiveness.

**Strengths:**

S1:The training-free nature of the approach gives the method a clear practical edge, avoiding the high cost of fine-tuning large models on domain-specific data.
S2：The finding that lightweight fused models can rival their heavyweight counterparts is valuable for resource limited scenarios.

**Weaknesses:**

W1:The key techniques include weighted fusion of image-text embeddings, using VLMs for caption generation, which is not novel. The contribution is primarily an engineering combination of existing methods applied to the food domain.
W2:The evaluation uses only small subsets (13K and 15K samples) from the MetaFood Challenge datasets, which raises questions about generalization. No evaluation on other well-known food datasets (e.g., Food-101, Recipe1M) is provided.
W3: In Equation 8, the weights (w_img=0.7, w_text=0.3) are set without ablation/parameter study. And in Equation 3, different weights are used for bi-directional fusion (w_img=0.3, w_text=0.7) without clear justification.
W4: A critical yet unaddressed limitation is that the framework invokes a full VLM forward pass for every single query image, an operation that carries non-trivial latency and GPU-hour expense. No empirical analysis is provided.
W5:  There is no comparison results with the cited training-free methods (s PDV Tursun et al. (2025), TF-ZS-CIR Wu et al. (2025)) in related works.

**Questions:**

D1: Could you clarify how the fusion weights were set? Was a grid search conducted, or was another tuning strategy employed? How sensitive are the retrieval results to small changes in these hyperparameters? Finally, what motivates the choice of different weight values for uni-directional versus bi-directional fusion—does the shift reflect a fundamental difference in how each pathway contributes to the combined representation?
D2：How does the pipeline behave when the VLM hallucinates an incorrect caption—e.g., mislabeling “green bell pepper” as “cucumber” or omitting a key ingredient?  And how to deal with some potential biases in VLM-generated descriptions?
D3:In Equation 11, what theoretical or empirical motivation led you to adopt max-pooling rather than an average or learnable weighted combination?

---

### Meta-Review · Area_Chair_4e1w · 2026-01-05

**Summary:**

The submission received unanimous reject ratings. Reviewers consistently identified significant weaknesses: limited technical novelty (relying on basic weighted fusion), insufficient evaluation (restricted to small subsets of a single dataset), and a lack of comparison to relevant training-free baselines.

**Reviewer Concerns:**

The authors did not submit a rebuttal. Consequently, all critical issues remain unaddressed.

**Reviewer Scores:**

Given the complete absence of author engagement, the reviewers would firmly maintain their initial reject ratings.

---

### Decision · Program_Chairs · 2026-01-26

Reject